# Participatory Learning and Co-Design for Sustainable Rural Living, Supporting the Revival of Indigenous Values and Community Resiliency in Sabrang Village, Indonesia

**Lira Anindita Utami [1,\*], Alex M. Lechner [2], Eka Permanasari [2], Pandu Purwandaru [1] and Deny Tri Ardianto [1]**

[1] Faculty of Fine Art and Design, Sebelas Maret University, Surakarta City 57126, Indonesia
[2] Urban Transformations Hub, Monash University Indonesia, BSD City, Tangerang Selatan 15345, Indonesia
\* Correspondence: lyra.utami@staff.uns.ac.id

**Abstract:** Industrialization and urbanization have affected Indonesia's rural communities and farming culture, which were once integral parts of its ecological system. This paper presents a participatory co-design approach based on the local and traditional learning philosophy of *niteni* to support sustainable development. The participatory co-design approach encouraged collaboration between marginalized communities, government bodies, and a multidisciplinary academic team. Through this lens, interviews, forums, and an ethnographic study were undertaken in order to acquire data and information for idea generation and planning. Firstly, eight *niteni* themes were identified, including the environment and ecosystems, traditional values and farming culture, crafting skills, manufacturing, and the local economy. Building on an understanding of the challenges associated with each of these themes, we identified future development priorities. A key action identified was the reintroduction of traditional farming, in particular the planting of local rice varieties and the local tradition of *Pranatamangsa*, which promote human–nature connections such as farming activities and rituals following natural seasonal cycles. Finally, design approaches were used to revive the local rice farming tradition (Rojolele Delanggu), including product branding and packaging designs to support regional identity. The paper concludes that the inclusion of design thinking in a sustainable development strategy based on cultural specificity can increase participation and support traditional indigenous practices and community resiliency.

**Keywords:** sustainable development; local wisdom; agriculture; participatory design; co-design

## 1. Introduction

Indonesia's rural communities and agricultural systems, as well as the relationship between villages and cities, have been impacted by industrialization and urbanization [1,2]. Since the Green Revolution in 1968, traditional farming methods in Indonesia's Javanese rural farming community have been transformed using more efficient, mechanized, and highly modernized approaches [3,4]. The farming culture that was once an integral component of the environment, securing biodiversity [5] and materializing collective beliefs and values based on traditional spirituality and philosophy, has gradually diminished due to the increase in commercial capacity and modern living [6,7].

For many, the wisdom of traditional farming culture, which once benefited from its coexistence with biodiversity, thoughtful natural resource management, and a mindful societal education, is less attractive and outdated compared to modern materialistic lifestyles. This transformation was further accelerated by Indonesia's Green Revolution policies, which favored the appointment of outside officials to communities, implementing policies and subsidies that enforce regional development agendas and technical efficiency in farming [8]. Furthermore, the policies related to urbanization have exacerbated the divide between villages and cities, resulting in the unequal economic growth that influenced

the Great Migration and has left rural areas exploited and underdeveloped [9]. These changes have also transformed kinship-centered agrarian communities into industrial-like agriculture with bureaucratic administrative systems and exploitative top-down relationships [10,11].

In rural communities, there is a need to pursue a different ideal and approach to design practices that can bring positive and sustainable social changes so as to support existing cultural practices and reduce environmental impacts. Designs for rural wellbeing and livelihoods need to account for a range of sustainability challenges while being sensitive to existing traditions and providing a voice for the community. Participatory co-design approaches based on local and traditional learning philosophy can support sustainable development. Participatory design approaches are carried out by learning about life in the field and becoming familiar with local wisdom, with guidance from the local community. Active participation triggers information exchange and co-produced knowledge through a range of activities [12,13]. Design principles, concepts, and their implementation should therefore be generated from local ideas and inspired by traditional culture.

This paper investigates the extent to which a participatory and co-design approaches with a conceptual learning and design framework can be manifested with the community. This research, conducted in Sabrang Village, Indonesia, aimed to generate awareness of sustainable living by reviving agricultural educational and cultural possibilities. We started by introducing a participatory design approach based on the ethos and knowledge systems of the local community. Then, using participatory learning and action techniques in participation with multiple stakeholders from the farming community, local government, academia, and students, we conducted field activities focused on ethnographic activities for the collection of data and information, material and cultural studies, and the mapping of local treasures for idea generation and planning. From these findings, we applied the notion of *design for living* and *design culture* as a design precept and a learning technique based on local culture in order to enhance sustainable development practices in Sabrang Village and offer recommendations for sustainable development pathways.

## 2. Review of the Co-Design Approach and Participatory Learning and Action Framework

Designers need to investigate opportunities to aid in social transformation not only in terms of their political implications, but also to support social change, potentially in combination with participatory educational approaches [14]. In many regions across the world, such as Indonesia, design education still employs a classical approach, where the user is the passive object of study [15–18]. In comparison, in the contemporary design discipline, designers must not only comprehend the precept and praxis of the participation design but also facilitate concrete planning and space-making for the co-creation of mutual learning and knowledge [19–22], encouraging new habits that lead to social change [23–25], recognizing regenerative potential in order to mediate the ethical advancement of both human civilization and the natural environment [26,27], recognize local wisdom, traditional technology, and biocultural heritage [28–34], and address the inconveniently absent relationship between villages and cities [35]. This new participatory design approach seeks to include and actively involve stakeholders, particularly future users or benefit recipients, as design partners or co-designers [36,37].

The term "participatory learning in design" draws upon the importance of reflective observation and actions of documentation for the purpose of knowledge acquisition through experience, as "the process of learning from experience is ubiquitous, present in human activity everywhere all the time" [38]. As the participatory learning in design process employs an ethnographic approach in order to understand the native point-of-view [39], it calls for involved participants to situate their five senses in the field in order to collect information about ways of living, ecology, historical records, cultural activities, religious rituals, food, and material culture, with a view to defining which activities can

be factored into a design to support the sustainability of the life of the impacted community. Participatory design should transcend cultural and professional barriers, supporting collective decision-making, and enable everyone to participate effectively in a variety of ways in the formulation of all decisions that affect them [40].

Social innovation by design, such as participatory design, refers to research and practices conducted by professional designers that emphasize the empowerment of marginalized groups of people by including them in the processes of design. Participatory design is a value-centered design approach [41], as it focuses on the facilitation of marginalized groups through participation [42–44], especially in the public sector, with the help of participatory learning and action processes [45]. Participatory design is often used by designers who specialize in regional development and community empowerment in order to examine a variety of socio-economic conditions, such as politics, ethos, belief systems, and emotional states that may be connected to ecological and environmental conditions, which can inform the design process. Sustainability transformations, which are often highlighted in any regional development or community empowerment study, also require a participatory design in order to be inclusive and attentive to the needs of non-humans [23,25]. Despite the evolution and diversification of participatory design approaches to addressing different aspects of inclusion, the anthropocentric value system remains dominant [46]. To tackle this, a conceptual design framework that addresses the needs of both human and non-human stakeholders should draw from the ethos and knowledge systems of the local community.

To design is to create a way of living. Therefore, design is at the heart of sustainable regional development [30,47,48], which should consider traditional ecological knowledge (TEK) [32]. In the design for living, the context of the design refers not only to technology or the practice of making everyday utilitarian objects but also to plans and culture-generating actions aiming to support people in their surroundings, activities, and communities, which contribute to the fulfillment of basic needs, such as a healthy environment for food security, economy, dwellings, socio-educational activity, or any creative pursuit. The locality and cultural specificity of a region and its indigenous community can also be identified by understanding the many factors that foster local potential, including humans and non-humans who dwell in the area and their relationships with stakeholders and traditional values, in order to encourage inclusive, pedagogical, and transformative design practices. The appropriation of local TEK and other cultural specificity in a region, in the context of design, can emphasize the traditionality of a region. This can promote not only the sustainability of co-design activities but also new possibilities, such as local tourism and the revitalization of regional identity, in order to increase the sense of attachment of the villagers to, and their satisfaction with, their own land, livelihoods, and tradition, therefore enhancing local pride as a method of maintaining participation in the protection and preservation of their local heritage [49].

Ezio Manzini [50] described the important aspects of social design as openness, connectedness, locality, familiarity, and responsibility, which can be achieved through small-scale projects aimed towards social innovation to increase collective potentialities. Manzini and Nigel Cross [14] emphasized the need for design practices that develop collaborative forms of engagement within the design process, where common people, citizens, local communities, and professionals from different disciplines can participate in decision-making, exchange knowledge deriving from diverse experiences, and gain insights into the intricate relationships and interconnectedness of things. The design approach encourages designers and non-designers alike to rethink the idea of local development. It often involves multiple stakeholders acting as a common good that enhances a sense of place, heritage, and sustainable knowledge and practices. In this sense, it addresses the questions of how collaboration, in a multi-sector and multi-actor context, is significant for the social dynamics and powerplay between the internal/external stakeholders and how this dynamic can foster proactive ability in the development process, or if the process

yields another top-down or egalitarian partnership and, indeed, if an outside perspective can help to reverse any negative trends that occur at the local level [34].

Participatory learning and action methods are employed in the processes of the design for living in order to encourage sense-making through participation in specific spaces and communities and practices that facilitate the learning of all participants by grasping their experience and transforming it. A foremost principle of these methods is the co-creation of knowledge and the process of its acquisition through experience that, at first, occurs peripherally, but then gradually increases in terms of complexity and engagement. People from outside the community, mostly designers, students, and other experts who position themselves as learners, expose themselves to subjective feelings and emotional conditions that are created by in situ interactions. In turn, this shapes the ways of thinking and judgment and influences collective action. In addition, the co-creation of knowledge and acquisition of experiential knowledge can serve to subjectify the narratives of the design process. The existing local narratives, such as prevalent myths and folklore that construct the ethnographic images which are grounded in the local community's activity and material culture, can be enhanced by external participants and, as a result, local identity can be reproduced based on experienced and historical memory [51]. In this sense, the production of the traditional and the local identity, through the co-production of ethnographic images by internal and external participants, is a continuous process of creativity and adjustment [52]. The recreated ethnographic images act as the raw materials that can transform any services and/or products yielded by the participatory process into culturally meaningful products that have value beyond their use or exchange value or be factored into a branding strategy [53].

The approach of participatory design is not limited to Western scholarship. In Japan, the cathartic post-war reformation and environmental devastation caused by nation-wide industrialization, driven by capitalism and over-consumption, exacerbated social issues and prompted Japanese designers to reconsider the value of design and re-orient their approaches [45,54–57]. Research approaches transitioned from methods that were heavily oriented towards the Western tradition of industry-led design to one based on reigniting traditional wisdom, values, and their influence, oriented towards the preservation of symbiotic relationships between humans and nature, which are core tenets of Japanese living. Similar to the Scandinavian design tradition, which focuses on the democratization of design in everyday society through the dialectics of tradition and transcendence in order to address the tension between *what is* and *what could be* [58], Japanese design developed a participatory approach heavily inspired by the traditional ethos and aesthetics of craftsmen who create the culture that embodies the philosophies of Shintoism and Buddhism [59]. Japanese design was directed towards a more holistic approach so as to address environmental and socio-spiritual issues to design and implement community/regional development models for promoting social acceptability, economic resiliency, and environmental soundness. For Japanese design scholars and practitioners, this modern design practice underwent an indigenization process and became a common folk activity, resonating with older methods of object-making *(mono-dzukuri),* tool-using, and craft traditions of folk-culture [55,59]. In the Japanese local terminology, the real meaning of design comes from the word *ishou* (Jp. 意匠, merging the two sinographs 意 (i), meaning will or intention (produced in the immaterial heart), and 匠 (shou), meaning craft-making) [45]. Thus, design can be defined as a form of intention, reflecting the heart of a designer, who uses his hands, skills, and tools in order to materialize intended, perceivable objects.

Many design practices in Japan are based on traditional cultural values and deal with identity-making, community-building, and relation-making. In the Japanese context, design is a creative engagement activity that not only considers the creation/use and embellishment of objects or the application of cutting-edge technology but also, most importantly, concerns the mind–heart–body interaction in the embodiment of intentions vis-a-vis the purpose and the lives that it will affect. Following this perspective, the design process needs to incorporate bi-directional experiential learning, the generation of

meaning, the formation of ideas, and decision-making activities, in addition to form-making, which allows a group or individuals to build new capacities, shape public understanding, and change their lifeworld. In this way of thinking, design is a way of addressing social issues, where certain subjects are co-produced in a given space and time through processes, participation, and the utilization of the designed objects (systems, services, or products) [60–62].

Within the sphere of the Japanese traditional education system, allegorical examples drawn from everyday life, agriculture, or expressions of nature influence many learning precepts (Jp. *iinarawashi*) designed to pass on wisdom, philosophy, or complex ideas in simple and memorable ways. Among the principles of the design for living, the design culture generated by the Japanese design culture lab, the precept *to learn about life in the field* (Jp. *no ni dete seikatsu wo manabu*) is derived from an ethos through which the method and tools of participatory learning and action are developed. With this precept in mind, designers/students are encouraged to actively engage with the people and natural environment, involving themselves in living scenarios in order to learn the ways of life, history, local wisdom, and materials, so as to collect insights and knowledge for the development of participatory techniques.

In this project, we adopted Japanese design values within the context of the guiding principles of participatory design, which are primarily concerned with facilitating democratic practices, situation-based actions, and mutual learning through activities and workshops. Mutual learning is acquired primarily through exposure to phenomena, direct interactions, and experience of the living spaces of people who are engaged in activities with one another and/or with the lived environment (e.g., nature and life forms), albeit unconsciously. In this transdisciplinary design practice, the experiences of those involved in the design may also be valuable, as knowledge is co-created to inform and influence the outcome of the design, which often takes the form of a service, artifact, or intervention [63]. The recognition of this whole environment, both internally and externally, is also considered to be important [64], since it creates a space for acknowledging the importance of traditional knowledge and value systems.

## 3. Methods

### 3.1. Case Study Description and Background of Sabrang Village, Indonesia

The participatory design approach described above was applied to the study of community development in the small village of Sabrang, located in Delanggu District, Klaten Regency, Central Java (Figure 1). Delanggu District is strategically located between the city of Surakarta and the Special Region of Yogyakarta. The area is known nationally as a potential rice-producing area. The sub-district area, which faces Mount Merapi and houses the Cokro springs, is a fertile and productive land used for rice farming and plantations. With the support of its geography, Delanggu has several local rice varieties, such as Rojolele Delanggu and Mentik Wangi rice. Within the Klaten Regency, Delanggu District is included among the top 10 rice-producing areas, yielding up to 20,395 tons. However, since the Green Revolution mandated fast-growing rice farming, thus encouraging the use of chemical fertilizers and pesticides as well as waste accumulation in the land and rivers, the once fertile soil has gradually lost its nutrients, affecting the crop production quality. This ecological degradation has largely been caused by socioeconomic issues associated with the extinction of traditional farming practices. Modern mechanized farming techniques and fast-growing rice strains have impacted cultural and ecological patterns that were once embodied in environmental ethics through farming rituals and festivals. Modern farming approaches include the adoption of a market-based system centered on profit and labor efficiency.

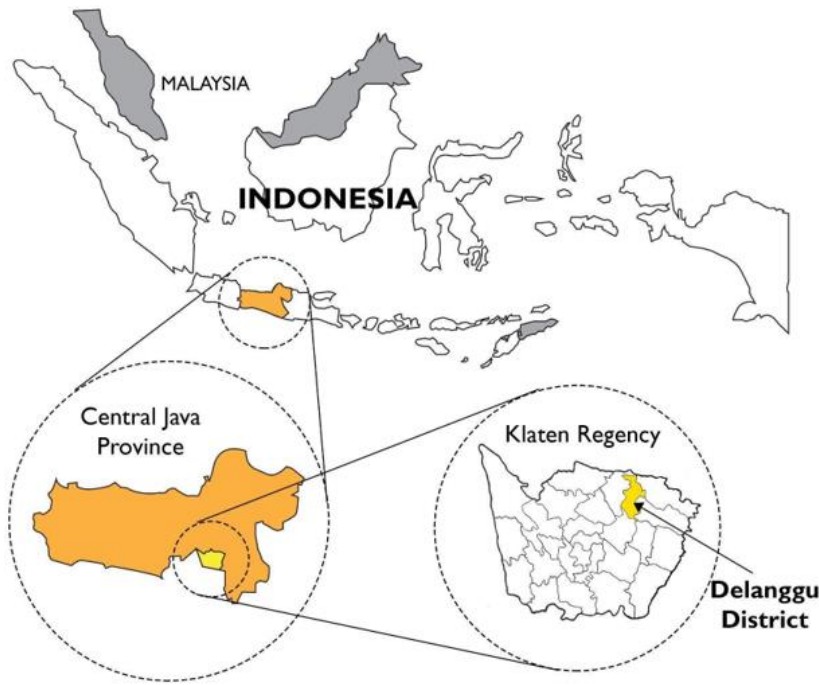

**Figure 1.** The location of the study area: Delanggu district, Central Java, Indonesia.

In the indigenous Javanese education system, *niteni* is the act of observing, signifying, recording, and relating to phenomena associated with nature and wildlife in order to determine the processes of farming and planting. For the Javanese people, *niteni* represents the local wisdom that created their traditional knowledge system, the *Pranatamangsa.* This system established seasonal guidelines for various social and economic activities centered on farming and planting, which have been lost across Indonesia. When considering the participatory learning and action method for the Sabrang regional development project, we recognized *niteni* as the Javanese version of the Japanese experiential learning system of *no ni dete seikatsu wo manabu* (to learn about life in the field). This system seeks to discover the learning potential that may contribute to the curricula of the participatory design—in other words, the act of learning from indigenous perspectives.

*Niteni* comes from the Javanese word *titeni*, which means to observe and to sign, and in turn comes from the word *titen,* an attitude required for learning that means (being) careful, acute, and thorough [65–68]. Based on this traditional way of learning, all human agricultural activity is the culmination of decisions and planning generated from lifelong learning, adaptions to nature and wildlife, and the formation of living skills, including object-making. The precept of *niteni* gives birth to a comprehensive agricultural time system based on natural cycles, recorded in local proverbs, interpretations, and procedural knowledge called the *Pranatamangsa*. While *niteni*, itself, is an indigenous learning technique developed by widespread farming and sea-faring people across Java, *Pranatamangsa* represents the indigenous knowledge system observed in traditional Javanese farming communities since the 19th century [69]. For these farmers, *niteni* and *Pranatamangsa* are evidence of the ways in which the natural world, as well as its biodiversity, climate, and wildlife activity, intermingle and co-influence human activity in order to create a sustainable living environment. From their perspective of traditional ecological knowledge, all human activities must fit into this already established ecosystem and form symbiotic relationships with other beings in traditional landscapes.

*3.2. Application of the Participatory Design Approach*

The co-design process implemented in this study was a participatory learning approach that involved multiple stakeholders (academics from faculties of art and design, agriculture, engineering, and urban design, as well as students, village inhabitants, and local governments) and ensured that the process was experienced by all participants [14,40,45]. Our participatory design approach utilized ethnographic works, emphasizing localized design thinking combined with culture-generated practices as a means of social innovation [50]. The participatory design framework was adapted from the Japanese design tradition, which is focused on the participatory learning approach and craftsmanship and can be traced back to the Scandinavian design tradition, in line with a value-oriented design [58] for social innovation. It employs participatory learning and action as methods and associated tools for generating collaborative movement between the project participants and sustainable regional development, which can be used to tackle social/economic/ecological issues and pave the way for community resilience [45,48,59].

The participants, including the research and design team, consisted of individuals with diverse educational and experiential backgrounds, elucidating the different interests and knowledge that must be reflected in the field. A total of 58 individuals were involved in these activities: 28 from outside the community, 24 from the community, and 6 key persons from the farmers' associations and government collective. From both farmers' associations (Sedyo Makmur and Sedyo Mulyo), a collective of 8 farmers were consulted in routine interviews. A breakdown of the participants by their activities is found in Table 1. The level of involvement varied between participants. For example, some of the participants, such as the co-authors, contributed to nearly every activity, while some members of the community only participated in one of the activities. To ensure the level of community engagement and participation, this research also analyzed the farmers' satisfaction through the use of reflective interviews. Design principles and concepts were used to construct ethnographic narratives generated from local ideas and traditional culture and to co-design the activities (Figure 2).

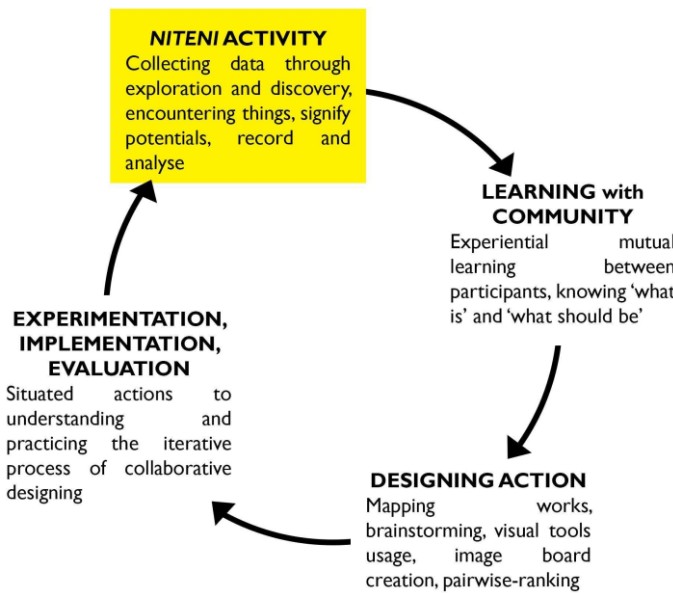

**Figure 2.** The flow and iterative process of co-design in a spiral model.

**Table 1.** Summary of the five activities and their methods, sample size, participants, duration and date.

| | Activities | | | | |
|---|---|---|---|---|---|
| | **Initiation Meeting** | *Niteni* **Treasure Mapping** | **Idea Generation and Future Priorities** | **Craft Design Workshops** | **Project Reflection and Evaluation** |
| **Methods** | Focus group discussions (FGDs) and learning with the community | Observations, taking notes and pictures, interviews, treasure mapping, and FGDs | Potential future development idea mapping, pairwise ranking, and FGDs | Priority-based activity design, craft workshops, logo design, storytelling, rapid prototyping, social media establishment, and FGDs | Short survey and FGDs |
| **Sample size and participants' description** | A total of 6 key persons from farmers' associations, 2 government collectives, 6 student representatives, and 2 lecturers | A total of 58 individuals, including 28 from outside the community, 24 from the community, and 6 key persons from the farmers' associations and government collective | A total of 58 individuals, including 28 from outside the community, 24 from the community, and 6 key persons from the farmers' associations and government collective | A total of 6 key individuals from the farmers' associations, including 2 from the government collective, 6 student representatives, and 3 lecturers | A total of 12 farmers from the farmers' associations |
| **Duration and date** | 2 December 2020 for 3 h to 4 h December 2020 for 3 h | 15 January 2021–22 January 2021, 10 March 2021–17 March 2021, 13 May 2021–20 May 2021, 3 July 2021–10 July 2021, 3 h each day | 3 August 2021–10 August 2021, 3 h each day | 13 August 2021–15 August 2021, 3 h each day | 17 August 2021 for 5 h |

*3.3. Design Process and Participatory Learning and Action through Niteni*

The participatory design project undertaken in the Delanggu farming community in Sabrang was initiated in 2020 in the wake of the COVID-19 pandemic and is currently still being executed in a restricted fashion, due to the policy of limited movement. The authors used co-design in order to identify and implement activities, but also for the anticipation of knowledge gaps and psychological issues, such as over-dependence and low confidence, as a result of differences in socio-cultural and knowledge between the internal actors. There were 5 activities undertaken for this project, which included a range of participants, sample sizes, and types of engagement (Table 1), namely: (1) an initiation meeting, (2) *niteni* treasure mapping, (3) idea generation and future priorities, (4) craft design workshops, and (5) project reflection and evaluation. These five activities were undertaken using the *niteni* learning framework and a participatory design approach, which encourages the participation of farmers, the government, and village residents.

### 3.3.1. Initiation Meeting

The project was initiated after several meetings and discussions (focus group discussions) with the local farming community and regional government, adopting a participatory learning and action design approach to account for human and non-human stakeholders co-existing in the agricultural environment. Since one of the targets of the co-design process was to enhance harmonious living and collaboration between stakeholders, the initiation meeting focused on strategizing methods of organically generating local wisdom. One way to achieve this was by encouraging local human stakeholders, such as farmers, residents, and local government officials, to actively participate as the main actors, while external stakeholders, consisting of multi-disciplinary experts positioning themselves as part of the community, provided an external lens and offered different perspectives in order to generate insights. The goal of the initiation meeting was to trigger endogenous development and learning from each other, and to identify regional potentials, such as wildlife, culture, traditional values, history, architecture, artifacts, and food culture, as sources of inspiration. The initiation meeting introduced and co-developed the ultimate goal of this approach, which was to create a community whose social climate, practices, and activities could support economic activity, social acceptability, and environmental sustainability. Since the agricultural policies imposed by the regional government are a key driver of many of the changing techniques used to educate government officials about this co-design approach, they also were included as key stakeholders.

### 3.3.2. *Niteni* Treasure Mapping

The *niteni* field exploration was the second step (Figure 2). This activity emphasized each participating individual's perception and interpretation of objects or things as treasure (Figure 3). In the initial step of *niteni*, perspectives were gathered by applying the method of field exploration, called treasure mapping (Figure 3), in which a small party of academics, students, and representatives from the community learned to (re)recognize forms of *niteni* related to people's habits, objects of bricolage, and farming activities, i.e., to identify traditional knowledge, activities, values, etc. Participants engaged with the community and local environment so as to engender the activity of exploration and discovery through the formulation of praxis-generating keywords: encounter, signify, record, and analyze. The cognitive capacity and judgment of aesthetic experiences, formed through the body's engagement in repetitive endeavors and reinforced by feelings and reflection [70], and processes of improvisation with the social, material, and experiential resources available are perceptively acquired [71].

Here, *niteni*, as the conceptual framework of experiential participatory learning and knowledge production, is applied to the initial activity of field exploration, often called treasure mapping, which seeks to recognize, collect, and map the treasures, including regional land, farming traditions, and people, as potential bases of knowledge and actions for the co-design project. In practice, *niteni* brings about the memory of embodied/embedded perceptions of the environment and the ways that individuals manage and relate to them [72]. In field activities, the precept induces the students/designers to position the local community as teachers and the area of exploration as a place to gather signs, signifiers, and relations. With the satellite imaging map in their hands, participants walked through the village, taking notes and documenting matters and objects of interests.

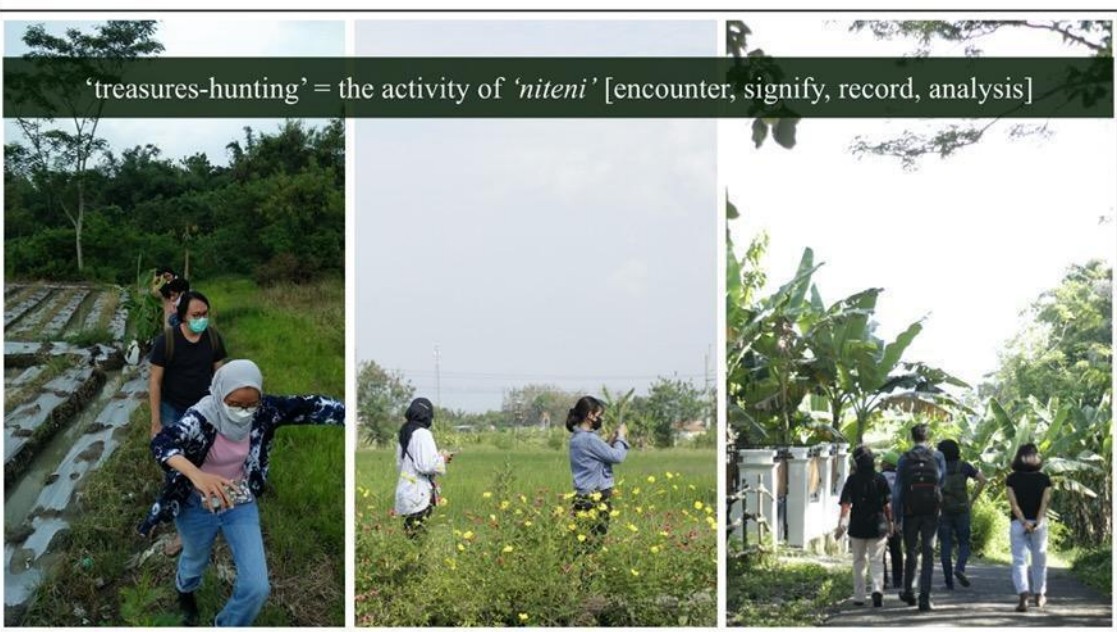

**Figure 3.** *Niteni* activity is generated and formulated to assist in the learning process of co-design.

While participants from outside the community may recognize and value traditional agricultural systems, on the other hand, for the local people, being exposed to routine living scenarios and natural sceneries may hinder them from recognizing the potentiality of the things around them, such that they may no longer be sensitive to these existing treasures. However, the presence of external stakeholders may encourage them to observe local potentials and voice opinions. As part of the process, the local community learns to (re)recognize existing values while acting as teachers and providing various narratives in response to the surveyors' inquiries. Insightful outcomes may be achieved through fresh eyes, but those eyes must also be educated and intelligent [73].

### 3.3.3. Idea Generation and Future Priorities

In this step, the participants discussed the results of the first phase the *niteni* activity (2021), in offline and online informal forums in order to generate future development priorities. Potential future development idea mapping was generated through the use of guiding questions in order to categorize the themes and their associated activities uncovered by the first activity (Figure 4). These tools for idea generation included area maps, used to characterize topics from a system perspective, and digital tools, used to form pairwise rankings and visualize prioritizations. These mapping and brainstorming tools are critical for ensuring that non-designer participants are included and heard during the idea generation and decision-making processes. These techniques also illuminate new ideas arising from the design process and assist the design/research team in gaining an understanding of complex socio-political issues, as well as power dynamics, within the community and their effects on traditional culture and ecology. These tools contribute to both the understanding of the current situation, the formation of short- and long-term community development plans, and the division of development priorities into several phases.

The final step in this activity was the application of a pairwise ranking method to determine a set of priority future development activities that the community were keen to develop going forward. In the pairwise ranking method, future development priorities are determined by focus group discussions. In the FGDs, the pairwise ranking method requires participants to compare two alternative development priorities one at a time and select which is more important. By comparing all the alternative pairings, development priorities can then be ranked from the most important to least based on the number of times that a development priority was preferred.

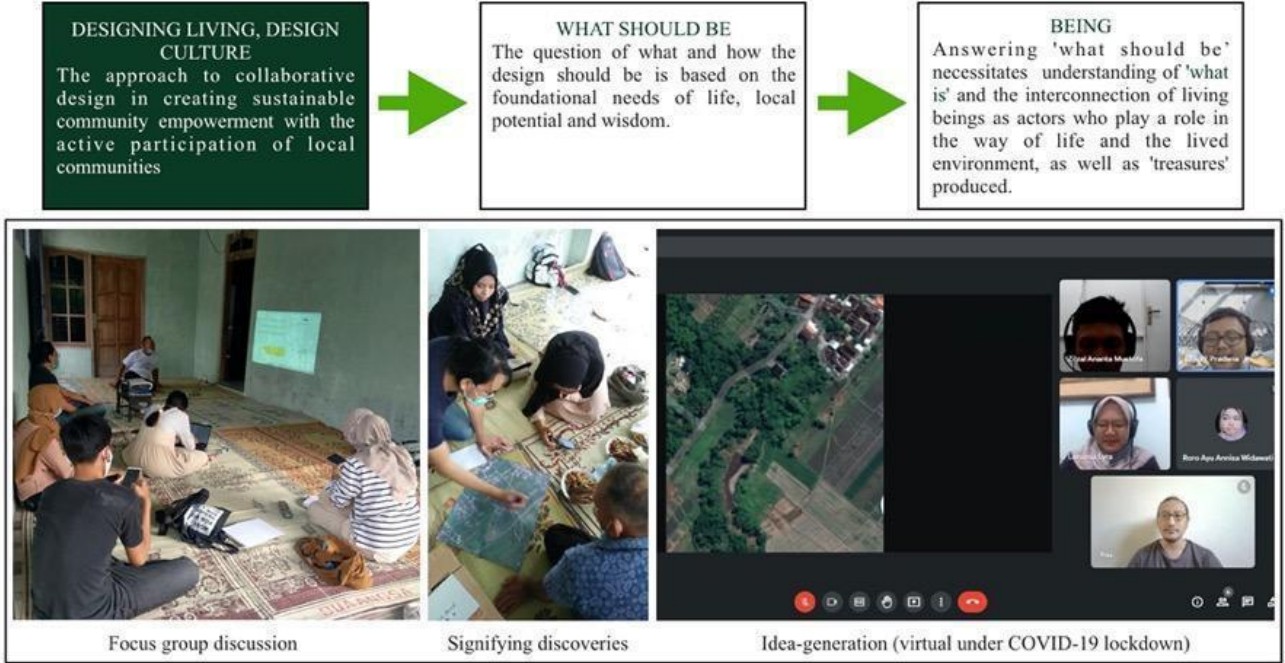

**Figure 4.** Approach of the design for living, design culture, and application in the Sabrang co-design project.

### 3.3.4. Product Design Workshops

In the product design step, all stakeholders were engaged in strategizing how to re-introduce traditional values through a product design workshop, based on the priorities identified in the previous step. This activity reaffirmed the status of community members, such as elders/farmers, as knowledge holders and the importance of traditional values. This step included taking a practical design perspective—that is, a priority-based activity design—that considered service and product designs and planning works which could be acted upon by the community. Like the previous steps, this step was conducted as a co-creation activity between the designers, students, and active community members. This step identified the services and products from Sabrang village that could be used to distinguish the region from other rice-producing regions in Central Java through the service and product design activities primarily formulated in the focus group discussions. These design activities included craft workshops, logo design, rapid prototyping for packaging, and storytelling. Furthermore, the designs created during this stage were promoted on social media (social media establishment), once again using a co-design approach in co-operation with the community.

### 3.3.5. Project Reflection

In the second year of the study, we included a stage of reflection about the project using a short quantitative social survey of the farmers' attitudes towards the co-design participatory approach applied in context and drawing on the perceptions of the participants in this project (Appendix A). This stage of reflection also considered the success of the activities in rebuilding an understanding of indigenous philosophies to support sustainable development.

## 4. Results

### 4.1. Initiation Meeting and Niteni Treasure Mapping

The participatory processes were co-developed organically by all participants with the overarching aims, goals, and objectives identified at the initiation meeting. In addition, at the initiation meeting, future activities, the involvement of stakeholders, and the practical design of the project were discussed while recognizing that there would be scope for change and that the project would be co-developed by all participants.

In the first step of the project, eight *niteni* themes were identified: (1) rice fields, (2) the environment and ecosystem, (3) activities, (4) livelihood and people, (5) traditional values and farming culture, (6) other public infrastructure, (7) crafting skills, manufacturing, and the economy, and (8) nature and scenery (Table 2). Through this simple engagement, everyone exchanged perspectives, learned to see things differently, and used tools to visualize ideas. These activities included the sharing of meals during discussion forums, taking photographs, and the use of drawings and storytelling during visual presentations. These activities created a more relaxed environment, in which participants and internal stakeholders engaged in expressive dialogue.

Moving beyond the treasure mapping activities, implemented through informal interviews and focus group discussions, a range of challenges associated with each of the eight *niteni* themes were identified (Table 2). We confirmed that farmers face issues related to modernized agricultural practices and the usage of various chemical agents that are destroying the natural nutrient cycle, depleting biodiversity, and yielding less favorable production rates—an issue facing agriculture across Indonesia. Other more specific issues related to farming in the region were also identified, such as water and land degradation caused by the accumulation of household waste and plastic waste, as well as the use of chemical pesticides. Issues regarding the lack of good economic models for the promotion of traditional farming were also identified.

Specifically, the assessment found that farmers have never had the opportunity to be self-sustaining in terms of the marketing of traditional crops due to their dependence on the central government and larger agricultural corporations. In terms of traditional values and cultural experience, they experienced the loss of the true taste of Delanggu as a result of their inability to plant the local rice, Rojolele Delanggu. The community identified a loss of the food culture that was born from this authentic rice variety. For the past thirty years, farmers have not planted local rice varieties, and various traditional activities have not been practiced for twenty-five years. There have been no farming activities aiming to promote local customs, since the community has had to rely on the government's agricultural agenda. The damage is threefold. Firstly, the community's social-spiritual needs are neglected, as there are no agricultural-based social platforms for the promotion of generational communal values. Secondly, there are no social gatherings aiming to prepare farming-associated festivals that promote the local customs and food culture. Thirdly, a decreasing sense of place, sense of ownership, and pride in the land was observed especially in the younger generation.

The loss of traditional practices has been exacerbated by the youth's lack of interest in farming. Furthermore, the youth association (Karang Taruna) no longer actively participates in village affairs, resulting in the loss of the sense of solidarity, namely *guyub* (comradeship) and *gotong royong* (collaboration and cooperativeness). Routine farming activities and top-down initiatives, without any development or accompaniment, have caused the farmers to lose their 'sense of growing the land' and connection to, or respect for, nature. Soil and water degradation is also considered as a cause of the bland taste of crops, such as vegetables and tubers. In conjunction, we found that the culture centered on ecological preservation, including the treatment of the land in pre-farming and post-harvest activities, as well as the maintenance of rice fields, the control of nature-supported pests, and the associated rituals, such as Wiwitan, Ngani-ngani, and Slametan, which created awareness of the symbiotic relationships, had been loss and/or remained unperformed.

The participants also identified climate change and its effects on *Pranatamangsa* as challenges. The traditional natural cycles could no longer be relied on by the farmers; however, they still found a place for the indigenous education of *niteni* by reflecting on the importance of nature and human interactions fostered through traditional practices. With the climate crisis worsening and the symbiotic culture disappearing, it was time to reconsider their approach to sustainability and innovation [32].

**Table 2.** Mapping of the local potential domains of *niteni* activities, including their problems and potential. These were identified by 20 students, 2 farmers, 2 design lecturers, and 1 engineering expert.

| | *Niteni* Activity Themes | Discoveries (Local Treasures) | Problems Founds | Resource Potential |
|---|---|---|---|---|
| 1 | Rice field area | • Around 4000 m² of the farming area and Polar Ijo<br>• Nearby small forests and rivers<br>• Railway that cuts through the field<br>• Bamboo traditional resting huts<br>• Flower plants and fruit trees (papaya, tomatoes, cassava, wild eggplants, ground cherries)<br>• Rice field paths (*Galengan sawah*)<br>• Irrigation of the Cokro River | • Household waste and plastic waste were found scattered around the rice-fields, on the paths, and in irrigation canals,<br>• Pests (snails, rats, hoppers, locusts, seed-eating birds)<br>• Huts were unmaintained<br>• Rice field paths were unmaintained and uneven<br>• Access to the river was difficult<br>• Dirt road to rice field<br>• Slippery road,<br>• Nearby waste-burning grounds<br>• Residue of chemical pesticide attached to plants or taken in by groundwater | • Repair of resting huts repair, their redesign, and construction<br>• Soil and water treatments<br>• Rejuvenation of rice field paths<br>• Creation of forest paths and river cleaning<br>• Revival of the local rice variety (Rojolele Delanggu)<br>• Semi-organic farming<br>• Organic fertilizers |
| 2 | Environment and ecosystem | • Small forests and bamboo forests, nearby river<br>• Natural predators (owls, herons, snakes, civets, bats, frogs, freshwater crabs)<br>• Edible snails, indigenous bird varieties (herons, estrildid finches)<br>• Shading trees (Muntingia calabura, teak and mango trees, hibiscus trees, breadfruit trees)<br>• Crop-yielding plants (rice, tubers, vegetables, herbs, fruits),<br>• Flowering plants (cosmos, Asian pigeonwings, refugias)<br>• Wild plants (groundcherry, Napier grass, cogon grass, common water hyacinth, nut grass) and edible wild plants (Limnocharis flava, taro, cosmos, Asian pigeonwings, Muntingia calabura, river tamarind, cogon grass) | • Household waste and plastic waste scattered in the river<br>• Access to small forests and rivers was difficult<br>• Scorching sunlight<br>• Abandoned rice fields near river<br>• Slippery forest paths<br>• Dull-colored and smelly river water<br>• Smelly irrigation canals | • Edu-ecotourism (forest-walk, river walk, food culture)<br>• Food crops and medicinal herbs for food security<br>• Natural pest control: owls, bats, Refugias, Javan mongoose, civets, snakes<br>• Nature-based water treatments and geotextiles for rice field paths.<br>• Natural dyes from wild plants and flowers<br>• Waste management system |

| # | Category | | | |
|---|---|---|---|---|
| 3 | Activities and livelihood | • Rice farming, crop gardening<br>• Establishment of small food stalls (Angkringan) and home-based grocery stores (Warung)<br>• Collective cooking<br>• Making fiber ropes (Gedeng/Lulup Waru, made from Hibiscus tiliaceus)<br>• Fishing, discussion, and Qur'an-reading gatherings<br>• Collecting woods and used cardboards<br>• Co-gardening and buying and selling crops in communal gardens | • Reduced cultural activities in which smaller children can participate<br>• Reduced youth activities related to the sustainment of local tradition<br>• Some littering was observed<br>• No specific educational activity concerning farming or the preservation of nature | • Promotion of local farming methods.<br>• Craft workshop and cultural activities<br>• Regional branding and marketing<br>• Making and selling of local foods<br>• Promotion of local materials through farming and craft development<br>• Revival of traditional games<br>• Activities stunting awareness and prevention |
| 4 | People | • Farmers' Association (GAPOKTAN Sedyo Makmur)<br>• Female Farmers' Association (KWT Sedyo Mulyo)<br>• Agriculture Service (PPL Dinas Pertanian)<br>• Youth Association (Karang Taruna)<br>• Schoolchildren of different educational levels | • Inactive youth organization<br>• Lack of interest in the inherited tradition or farming culture<br>• Risk of stunting | • Co-designing the revival of traditional games<br>• Implementation of participatory learning, with *niteni* as an extra-curricular activity and farming activity for empowering local education<br>• Co-designing disaster prevention education<br>• Cooperation in festivals related to food culture<br>• Cooperation in farming rituals<br>• Development of craft technology and skills |
| 5 | Traditional values and farming culture | • Planting ritual (Nyajeni),<br>• Harvest festival (Ngani-ani)<br>• Post-harvest festival (Wiwitan)<br>• Feasting (Slametan)<br>• Collective cooking<br>• Collaborative working in the lived environment (Gotong-Royong, Kerja Bakti)<br>• Traditional values and the building philosophy exhibited through hut-making and traditional architecture<br>• Some farmers still practice '*niteni*' to determine the times for farming and rituals.<br>• Local myth and stories | • A total of 35 years spent not farming the local rice grain resulted in the loss of local taste and degradation of food culture<br>• Loss of solidarity due to high competition in crop production<br>• Decreasing number of farmers/no regeneration of farmers<br>• Abandonment of traditional activities related to agriculture<br>• Abandonment of rituals related to the sustainable symbiotic relationship between humans and nature<br>• Abandonment of local farming to support a healthy ecosystem and biodiversity<br>• Loss of the traditional ecological knowledge system (Pranatamangsa) due to climate crisis | • Local rice grain (Rojolele Delanggu), the superior rice grain<br>• Traditional soil rejuvenation<br>• Revival of farming-related festivals and rituals (Nyajeni, Ngani-ngani, Wiwitan, Slametan).<br>• Revival of food culture (festival of the five senses)<br>• Natural pest control<br>• Farming-related skill enhancement<br>• Farming tool design<br>• Revival of traditional values (*Gotong Royong, Guyub*) |

| | | | | |
|---|---|---|---|---|
| | | | • Reduced sense of '*guyub*' (comradeship) | |
| 6 | Other public infrastructure | • Cemented roads<br>• Mosque<br>• Kindergarten<br>• Elementary school<br>• Junior high school<br>• Public cemetery<br>• Public spaces<br>• Communal garden (KWT Mojo Ayem)<br>• Communal security post (Ronda) | • Some unmaintained roads within the village<br>• No clear rules concerning garbage dump sites or waste management (some households still littering around the premise and within small forests nearby)<br>• Unmaintained communal garden and kitchen | • Refining road access to the village<br>• Mural-making activity<br>• Collective garbage bins or composting bins<br>• Rice granaries<br>• Straw-drying places and systems<br>• Communal kitchen<br>• Better street lighting<br>• Wayfinding signage |
| 7 | Crafting skills, manufacturing, and the economy | • Straw puppet making<br>• Fiber rope making<br>• Simple farming tools<br>• Home-based grocery stores (Warung)<br>• Food stalls (Angkringan)<br>• Repair workshops | • Loss of bamboo-based construction skills<br>• Degradation of crafting skills (straw-weaving, ropemaking)<br>• Degradation of traditional cooking skills | • Bamboo-based weaving and hut construction skill development<br>• Rojolele-straw-based craft development<br>• Fiber rope skill development<br>• Promotion of the local food and beverage (festival of the five-senses) |
| 8 | Nature and scenery | • Railway<br>• Small Forests,<br>• Rivers<br>• Rice fields<br>• Traditional huts and houses | • Wasted land near the railway and rivers<br>• No raised barrier near the railway (ditch can be seen)<br>• Traditional house structure not maintained | • Collective flower and vegetable gardens around rice fields and within villages<br>• Outdoor experiential learning spaces |

### 4.2. Idea Generation and Future Priorities

The first output from this activity was a set of development priorities generated by the participants. Participants with design backgrounds responded creatively to stimuli, which inspired non-designer members to expand their imaginations and participation in the idea generation process. Design students, who were also involved from the beginning, were especially eager to share what they had learned outside of university. Since agriculture-related knowledge has never been part of the formal school curriculum, they learned this information collectively through the project. In terms of co-design practice, the students learned the situated actions of participatory design in real-time, while the community learned how people from outside the village value the different domains of *niteni*, the treasures, which range from traditional values to nature and scenery. During this period, participants from outside the community stayed and developed relationships with the community and, in turn, acquired knowledge through experiential mutual learning. The outcome of this idea generation activity was a list of future development activities, which ranged from the branding of Rojelele Delanggu rice, logo design, packaging, and social-media outlets to ecotourism infrastructure (Table 3).

**Table 3.** Future development priorities (**A–H**) were evaluated through pairwise ranking. A value of 1 in the table indicates the development priority that was preferred for each pairing. The "total" column describes the total number of times that a development priority was preferred (i.e., given a 1). The rank was then derived from the total.

**Pairwise Ranking to determine development priorities through FGD:**

A. Branding of Rojolele rice, designing logo, packaging and social-media outlets
B. Development of Rojolele straw-based handicraft
C. Revitalization of farming-associated rituals and festival (Wiwitan, Ngani-ani, Slametan)
D. Food Culture (revival of Delanggu taste by Rojolele, local food and beverages)
E. Development of organic fertilizers
F. Re-cultication of local rice variety Rojolele Delanggu
G. Irrigation filters
H. Ecotourism infrastructure

|   | A | B | C | D | E | F | G | H | Total | Rank |
|---|---|---|---|---|---|---|---|---|-------|------|
| **A** | - | 1 | 0 | 0 | 1 | 0 | 1 | 1 | 4 | 4 |
| **B** | 0 | - | 0 | 0 | 1 | 0 | 1 | 1 | 3 | 5 |
| **C** | 1 | 1 | - | 1 | 1 | 0 | 1 | 1 | 6 | 2 |
| **D** | 1 | 1 | 0 | - | 1 | 0 | 1 | 1 | 5 | 3 |
| **E** | 0 | 0 | 0 | 0 | - | 0 | 1 | 1 | 2 | 6 |
| **F** | 1 | 1 | 1 | 1 | 1 | - | 1 | 1 | 7 | 1 |
| **G** | 0 | 0 | 0 | 0 | 0 | 0 | - | 0 | 0 | 8 |
| **H** | 0 | 0 | 0 | 0 | 0 | 0 | 1 | - | 1 | 7 |

Building on the future development priorities, pairwise rankings were used to rank the priorities based on their environmental soundness, indigenous values, and concerns about biodiversity preservation, socio-economic development, education, and food security. The top-ranked priority was the re-cultivation of local rice varieties (Rojolele Delanggu), which could be used as a promotion strategy for ecotourism. The second-highest-ranked priority was the revitalization of farming-associated rituals and festivals to restore the sense of gratitude, as well as *gotong-royong* and *guyub* (togetherness and comradeship). These performances celebrate the harvest, revive food culture, and encourage schoolchildren and women to familiarize themselves with the potential uses of rice straw by making straw puppets and crafts. This priority was also related to the third-highest priority, the revitalization of Delanggu food culture. While other priorities were not ranked as highly, they can also be considered as activities which support re-cultivation. These included the development of organic fertilizers, the addition of filters to the irrigation systems to reduce pollution in the water way, which focuses on engineering solutions, and the development of a new ecotourism infrastructure, which represents a type of activity that can be led by the community. Traditional farming methods based on *Pranata-mangsa* are centered on the cyclical observation of natural cycles, such as birds laying eggs and feeding hatchlings, and experiential learning is regarded as a way for people to understand this connection. Methods include backward-planting, soil-testing, the observation of rice plants from huts, and allowing local birds to hunt during the insect breeding season, because birds are considered as a natural form of pest control.

Follow-up interviews with the farmers and one village elder on the local traditions which could support priorities such as ecotourism confirmed the centrality of rice to many of these priorities. Rice is traditionally believed to be the embodiment of the rice goddess; hence, the farmers afford it respectful and gentle treatment in the form of a permission ritual and the cutting of the pinnacle using a specialized knife (*Ngani-ani*). Through this technique, longer and more pliant straw can be yielded, as the agricultural by-products can be used as raw materials for crafting and building. The villagers also stated that

traditional harvesting can only be performed by women, because traditional beliefs considered the rice plant to represent a pregnant goddess; hence, symbolically, women act as midwives by gently assisting the plants to deliver the rice. The ceremony continues, where *Wiwitan*, the embodiment of gratitude and appreciation of the harvest time, is performed at the rice fields and at the farmer's house, led by a female shaman. Following this, the *Slametan* festival takes place, with the entire village community cooking and eating together in order to celebrate a successful harvest. These local traditions are examples of potential activities that the project identified as requiring revival in order to re-affirm traditional ecological knowledge. In addition, the Rojolele Delanggu rice variety was considered to be well-suited to the land's geo-nutrients and was recognized as influencing a healthier farming culture that supports the natural ecosystem. Thus, its revival is crucial for the restoration of the nutrient cycle and the re-management of the land and irrigation systems.

### 4.3. Craft Design Workshops

The aim of the service and product design activity was to create design concepts to support the revival of local rice farming and rice varieties (Rojolele Delanggu), addressing the lack of promotion, marketing, and regional branding through farming associations, logo and packaging designs, food culture for the empowerment of regional identity, and potential craft product opportunities (Figure 5). Under the supervision of the farmers and village elders, describing folk stories related to rice, rice-farming activities, and the traditions and belief systems, external stakeholders (students and academics) designed product packaging. Using semantics, the identity of the rice-producing communities, and the value of rice in packaging illustrations, as well as written information concerning traditions, a farming method and nutritional content were created (Figure 5). This process was guided by the previous co-design activities.

Rojolele Delanggu rice was targeted for use in the premium market, using a marketing strategy that informs consumers about exactly what makes this local rice special and the identity of the community who produce it. In addition to designing the packaging, the design participants also developed strategies for social media platforms. The outputs and the media coverage resulted in a governmental agency contacting the community with the aim of registering the authentic rice variety as a regional specialty, and in interest from a company that sought to provide funding and to establish a facility in order to promote the taste of the rice and the community.

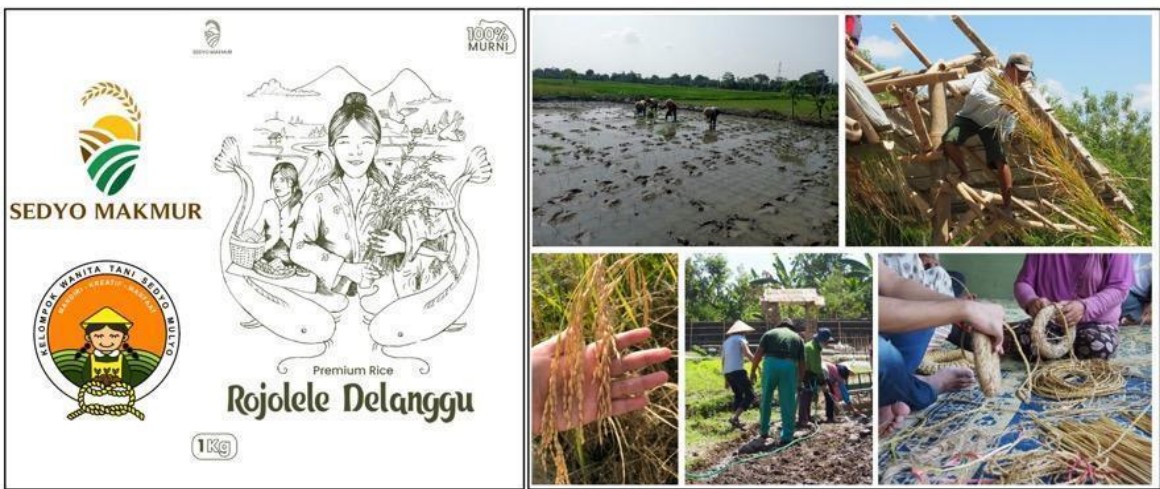

**Figure 5.** Service and product design activities, including the creation of a community logo, the rebranding of local rice, and the revitalization of traditional farming and craft making.

*4.4. Project Reflection and Evaluation*

The project in Sabrang village fostered a collaborative attitude among the stakeholders. Here, all involved members participated in the acquisition of new knowledge and knowledge sharing. Based on the evaluation of the 12 farmers' satisfaction levels, we found that the average level of satisfaction with the activities ranged from 3 to 5 on a 5-point Likert scale, where 5 is highly satisfied (Figure 6 and Appendix A). Meanwhile, based on our qualitative reflections, we believed that the activity was successful and observed that the designers and students learned from the farmers about the true potential of traditional farming, celebratory rituals, and the local food culture built on the esteemed taste of the indigenous rice variety. External stakeholders, who had never stepped into the muddy waters of rice fields, participated from the first step in clearing the field to the planting and care of the rice plants and the harvest.

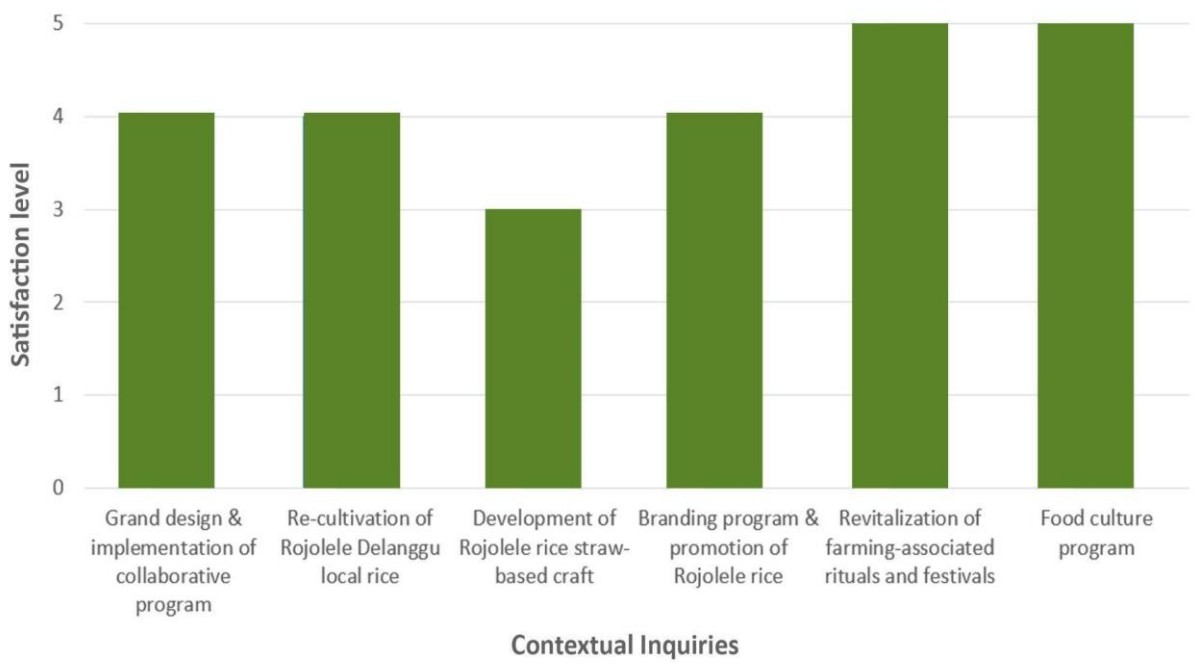

**Figure 6.** Average satisfaction level for 6 elements of the project's design outputs and activities.

**5. Discussion**

The participatory design approach applied in this study contributed to community activities on multiple levels, from the re-acclimation of traditional learning methods and the revitalization of the local rice variety and its associated ritualistic activities to the co-designing of services and products. For example, women farmers proposed their own activities, together with the university team, such as the preparation of a garden, initiating income-generating gardening activities, and the naming of their garden. The women farmers established their own group identity (Kelompok Wanita Tani Sedyo Mulyo, Women Farmers' Association, Sedyo Mulyo) and named a piece of land used for the activity (Mojo Ayem). Despite the differences among various participants, such joint activities undertaken with the university team demonstrated how the participatory approach provided a voice to all actors, including women, who typically have less opportunity to speak out. We found that the discussion forums and idea-generating visualization tools supported a range of perspectives. The process of *design for living* and *design culture* was based on mutual understanding, empathy, appreciation, and the experience of learning and action in the field. Drawing upon the precept of *niteni*, the farmers, local community, and university design students applied a diversified traditional ideology and praxis to the

discipline of design. Thus, this research intended to pave the way for a more inclusive design approach that can enrich the Indonesian design discipline. In particular, it can acclimate the pedagogical and experiential aspects of design for the purpose of the co-production of knowledge by both designer and non-designer participants. Following the six-step ladder of citizen participation mentioned by Arnstein (1969) [74], the process in Sabrang followed the method of partnership participation, where the community works together with the government sectors and other sectors.

This research demonstrated the possibility of the participatory design approach with a conceptual learning and design framework to be incorporated into a co-design process, undertaken in cooperation with the community in Sabrang village. *Niteni* provides mutual experiential learning in the forms of field exploration, discussion forums, and treasure mapping, with six design activity aspects aiming to revitalize Sabrang village's agricultural potential in multiple ways, including: (1) farming community empowerment through organic plantation and rice farming; (2) environmental soundness via the revitalization and conservation of local rice varieties (*Rojolele Delanggu*); (3) the rebranding of the Sabrang locality and craft making by exploring the community's potential to produce and manage agricultural by-products, such as straw taken as a raw material for production; (4) cultural festivals aiming to promote local entertainment and re-fortify social kinship; (5) the *niteni* practice of engendering learning activities by observing nature, people, and the activities of natural elements and wildlife in relation to the (re)recognition of potential treasures and knowledge; and (6) the generation of economic activities within the community by considering activities of the wider public, such as a farmers' markets, and establishing sites for co-design activities, such as educational eco-tourism.

Paving the way forward, we acknowledge that collaboration with the multiple stakeholders provides a good participatory design process which enabled the identification of experiential and mutual learning potentials [21,34,45,50]. However, in order to ensure that these activities are sustainable, the key leaders in the community and sectors which support the project must be maintained and reinforced. A fundamental challenge encountered in this project was the generation gap. Participants were mostly in their late 50s and mid-70s, and the immediate successor generation (who are in their productive years) was missing. Meanwhile, children in the community may inherit knowledge and be inspired by activities in their formative years participated. Maintaining their involvement is a challenge and a future opportunity, since, in this project, the children were excited to be involved in the participatory learning activities (especially in the ritualistic festivals, straw toy making, and traditional games) and were encouraged to participate further. Another challenge for the community was miscoordination and miscommunication with the government stakeholders concerning the management of funds. Finally, land and water management and adapting to climate change are crucial to the success of efforts to revive the authentic taste of the rice through ensuring the quality water and land. Only when the farmers harvested their first vegetables and tubers and tasted the blandness did they begin to recognize the connection between the conditions of the climate, nature, and the quality of their harvest.

The positive changes that were observed from the project included the attitude of the local government towards this project, with the government now acknowledging that there is economic value in revitalizing the almost-lost rice variety. The local people also learned to voice their opinions. Capacity building was also observed in the making of straw ropes, an initiative for creating natural ropes, aside from tree-bark ropes, with the intention of developing a craft movement. This work used to be the work of men, but now, rope making can be performed by women too. These participatory design processes are becoming embedded in the campus curriculum while also attracting the participation of more local people, such as local kindergarten children who, in this project, started to make toys out of straw, and revitalizing a traditional culture in which children play a role.

The participatory design practice used in the community-based development projects in Sabrang enabled all participants to learn, acquire, and exchange knowledge by

collectively discovering the indigenous potential to reflect on regional identity through experiential learning [19–22,70]. As indigenous potentials and values are recognized, they can be transformed into visually perceptible objects, services, or activities through design [32,45]. Participatory learning and actions increased confidence, especially among women farmers, who perceived themselves as uneducated. Farmers, in cooperation with the external stakeholders, consisting of design students and lecturers, learned to identify and recognize local potential. The experience of sensorial perception through situated actions, as developed by Asian design approaches such as *no ni dete seikatsu wo anabu* (Japan) and *niteni* (Java), provides a value-oriented framework of knowledge and understanding. These approaches echo design perspectives which are creative, imaginative, and investigative, growing out of real-life and hands-on situations. Drawing on data from participants engaging in *niteni* activities and design-led workshops, our study provided both an understanding of the local traditions and challenges, as well as actions that can provide design-based solutions.

The autoethnographic methods (narrative interviews, participant observation, and biographical methods) of the participatory design process applied in our study used narratives as a method of research in order to reconstruct local identity and preserve cultural heritage. These approaches have been applied in other rural regions, such as Poland, with Marcysiak and Prus (2017) [51] identifying the importance of the education of external participants (e.g., researchers, designers, and non-community participants) through participatory learning, especially attentive observation (similar to *niteni*) and socializing with the locals in order to gather stories or long-forgotten accounts of the community. These valuable sources of empirical data, based on ethnographic methods, are especially crucial when bringing people together with design-driven actions. Additionally, ethnographic images can be useful for creating a common narrative and strategy of territorial marketing [53] and promoting regional identity through service and product design, while attempting to boost the sense of pride of the local people in their own land.

The use of ethnographic heritage (including folklore and iconic scenery) in marketing approaches that celebrate Sabrang village life and its traditions (rice, rituals, craft, nature, and people) has attracted more customers of community-made products. This approach was inspired by Japan, which has area-specific types of rice produce along with attractive visual images and aesthetics that reflect the significance of the producing regions, as well as other locations and products around the world. For example, in the case of Bulgaria, the valorization of yogurt as both a traditional, "typical" Bulgarian food and an evidence-based health product was facilitated by sophisticated marketing forces and mythmaking [53]. The annual increase in Wiwitan ritual attendees (beginning in 2020) in Sabrang village and the purchase of rice whose profits go directly to the farmers' organizations are indicators of this expanding attractiveness of their community-made products. This was likely driven, in part, by the ways in which the packaging promoted an image of high-quality rice to consumers, aiding in the re-grounding of myths among the locals and consumers (and also influencing their visual aesthetics) and re-territorializing of their iconic identity (as the producers of Rojolele Delanggu), inspiring them, once again, to feel their attachment to the land and traditions and develop ethnographic images of the region and the rice farmers. This approach could be made more successful through official designation by the Indonesian government jurisdictions, as, for example, in the case of the Chinese village of Nalu, where the government has promoted the traditional village community life and products [49]. Sabrang could follow a similar path, since their marketing activities have attracted the interest of a governmental agency, and there may be a potential to authenticate Rojolele Delanggu rice as a traditional regional product (Sabrang, Delanggu).

Our study represents the first steps in an approach to the sustainable development of a whole region through the promotion of traditional values and produce. For example, Bindi et al., [34] empowered collaborations between multiple stakeholders in the area of Castel del Giudice in Italy, where social innovation was designed to manage natural and rural resources, as well as environmental heritage. The timely, yet consistent, practice of

organic agriculture of a locally grown apple variety led to the development of a local food plan, narrative, and new opportunities that provided meaningful actions for the people. This approach resulted in the creation of educational eco-tourism activities in the form of experiential tourism, aiming to diversify and expand local tourism and connect and create relationships between the service-providing community and tourists. A similar regional approach to diversifying the economic opportunities of the local community could be applied in Sabrang Village, which would promote the dynamic interplay between internal and external stakeholders in order to support heritage-based forms of political capacity-building, environmental sustainability, and ecotourism.

## 6. Conclusions

Discussion forums held between the research team, design team, and internal stakeholders facilitated the communication of real-world issues faced by the community and provided a platform whereby farmers could voice their opinions, consider a course of action, and cultivate and mediate present/future possibilities (the tension between what is and what should be) and adverse consequences. These participatory learning and *design for living* and *design culture* activities kickstarted the reintroduction of a food culture via the revitalization of Rojolele Delanggu and has had a positive impact on the preservation of the agricultural heritage, local rice production, and knowledge transfer. The farming community of Sabrang, the government officials, and external stakeholders have begun to envision and pave the way for the creation of new opportunities, such as experiential tourism, and the expansion of spaces of learning, aiming towards the revitalization of biodiversity and a circular economy and also to encourage participation from the next generation in the village.

**Author Contributions:** Conceptualization, all authors; methodology, L.A.U. and P.P.; formal analysis, L.A.U., P.P., E.P. and A.M.L.; investigation, L.A.U., P.P., E.P. and A.M.L.; resources, P.P. and D.T.A.; writing—original draft preparation, writing—review and editing all authors; project administration, P.P. and D.T.A.; funding acquisition, P.P. and D.T.A. All authors have read and agreed to the published version of the manuscript.

**Funding:** This research received funding from The International Collaboration Research, Universitas Sebelas Maret.

**Institutional Review Board Statement:** Ethical review and approval were granted internally from the Universitas Sebelas Maret and we ensured that participants involved in this research are kept anonymous.

**Informed Consent Statement:** Informed consent was obtained from all subjects involved in this study.

**Data Availability Statement:** The data presented in this study are available on request from the corresponding author. The data are not publicly available due to privacy and ethics considerations.

**Acknowledgments:** We would like to acknowledge Universitas Sebelas Maret for funding this research, the assistance of the community, particularly the Sedyo Makmur Farmers Group and Sedyo Mulyo Women Farmer Group, who participated in the research throughout this project, helping and providing data and sources of information to the researchers. We also extend our gratitude to the Department of Agriculture and Food Security of the Klaten Regency for supporting the development program, and the Head of Sabrang Village, who have provided extraordinary support to ensure the sustainability of this research.

**Conflicts of Interest:** The authors declare no conflict of interest.

**Appendix A. Evaluation of Farmers' Satisfaction with a Range of Activities**

| | Contextual Inquiries | Satisfaction Level and Results |
|---|---|---|
| 1 | Concerning the grand design and implementation of collaborative sustainable development programs | 1-2-3-4-⑤<br>Note: real contribution to the farmers' association (GAPOKTAN) is witnessed and appreciated. |
| 2 | Concerning re-cultivation program of local rice variety Rojolele Delanggu | 1-2-3-④-5<br>Recommendation: extend the farming fields.<br>Challenge: difficulty in finding human resources for farming. |
| 3 | Concerning the development of Rojolele straw-based craft | 1-2-③-4-5<br>Challenge: focus on, and continuity of, skill-building are still hard to attain for the participants (female farmers) because they are currently absorbed in the activities of re-cultivation and plantation programs. |
| 4 | Concerning the regional branding program and promotion of Rojolele Delanggu through the packaging design and social-media outlets | 1-2-3-④-5<br>Note: satisfied, people can access documentary video in the social-media platform.<br>Challenge: there is still a need to find suitable material for packaging, as current paper material is not strong enough. |
| 5 | Concerning the revitalization of farming-associated rituals and festivals | 1-2-3-4-⑤<br>Note: highly satisfied, because the abandoned traditional culture can be revived and practiced again. |
| 6 | Concerning the food culture program | 1-2-3-4-⑤<br>Note: highly satisfied, these activities are important for the internalization of traditional values (*Guyub*, *Gotong Royong*) and solidarity. Important aspects of education about traditional food and beverages, as well as cooking process, can be passed on to the next generation and promoted within the wider public. |

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
