# Peer review of "Participatory Learning and Co-Design for Sustainable Rural Living, Supporting the Revival of Indigenous Values and Community Resiliency in Sabrang Village, Indonesia"

_land, doi:10.3390/land11091597_

Round 1
Reviewer 1 Report
- The article addresses the very interesting and important topic of participatory design in the context of a community at the island of Java, Indonesia. Particularly the use of the Japanese inspired method and the focus on niteni make the article of interest for an international audience.
- However, the structure of the article could do with some signifanct improvements, e.g. by following the suggested structure of LAND. This would mean to have a separate discussion and conclusion section at the end of the paper. Part of what now is in section 3 at the end of the paper could be part of a discussion or conclusion section. Furthermore, a separate discussion and conclusion section would allow the authors to reflect on various topics and also to discuss what their case study contributes to the theoretical insights on participatory design (see also other points below for suggestions).
- In addition, a separate theory section after the introduction and before the methods section would allow to clearly present the theoretical angle of the paper, which is now partly done in the introduction and methods section.
- The introduction introduces the topic of the paper well. However, it a clear description of the knowledge gap the article is addressing, and/or the central research question is missing.
- The method section introduces the cases study extensively, providing much insight on the studied case. The section, though, misses a clear description of the research approach (which would be in the line of participatory (action) research if I have understood the paper correctly) as well as the sampling, data collection, and data analysis methods. This section could furthermore, be improved by presenting an anonymized overview of all participants, including their background, role, level of involvement, etc. In addition, all other methods used should be (briefly) described and included in the methods section. For example, on page 13, 14 ‘pair wise ranking’ is presented, this is missing in the methods section, this also accounts for the ‘measuring’ of the farmers satisfaction described on p. 17.
- The article addresses the topic of participatory design and presents an interesting case in the context of Indonesia. An important issue in any participatory processes is the level of participation that is reached (see e.g. the ladder of citizen participation presented by Arnstein (1969)). It would be interesting if the authors could reflect on the level of participation that was reached in (the different stages of) their case.
- Furthermore, since participation is a process over time, in which roles and relationships can change, it would be interesting to see if any (significant) changes could be observed in the studied case. Furthermore, as the article clearly addresses, learning takes place. The question then is who learned and what was the effect of this learning? Did (local) capacity building take place? Or did the process contribute to creating an enabling environment in which the developed ideas could be further developed? From this angle it is also important to present clearly who took the initiative for the participatory design process (did it come from within or without the local community), and how the processes ensured a connection with, and/or embedding in the existing (local, regional, national) institutional structures? It would be interesting to learn from the case on these kind of points in relation to what others have found on these points, e.g. in a separate discussion section.
- Figure 1 presents the case study location. Since LAND has an international audience and not all readers will be familiar with the geography of Indonesia, it would be good to add the location of the Delangu district in the context of Java/Indonesia.
- In Figure 2 the depicted arrows of the line and the (iterative) circles point in different directions, this is confusing
- The text needs extensive English editing, several grammatical errors and typos can be found throughout the text
- Page numbers should be added when direct quotes are used (e.g. on p. 1 line 80, line 91, p, 160 lines 329, 330)
Reviewer 2 Report
Dear Authors,
The manuscript in my opinion is interesting for readers of the journal. The authors had a good idea for a research project. The subject is relevant, the analytical methodologies are adequate, and the volume of data seems to be enough for publication. I have no hesitation in recommending publication after the following revision.
Not all articles cited by the authors are listed in the References section.
In the introduction, the authors briefly introduced the background and significance of the research. I consider it necessary to highlight the connection between the research objectives and the research hypotheses. The study's novelty and the highlighting of the main conclusions should also be emphasized. Besides, a description of the structure of the paper should be added.
It is recommended to shorten the content of the Method section, it seems too long. In my opinion, some of the content here is a literature review, and I suggest that the authors take out the relevant content as the second chapter, the literature review. I think this kind of revision will clarify the article's logic and give readers a better reading experience.
The Results and Discussion Sections should be separated.
The article is missing a Conclusion Section. Conclusions, limitations, and future perspectives of the study should be added here.
Reviewer 3 Report
Land (ISSN 2073-445X)
Manuscript ID: land-1869674
Type: Article
Number of Pages: 18
Title: Participatory design for sustainable living supporting experiential mutual learning, revival of indigenous values, and community resiliency in Sabrang Village Indonesia.
Dear Authors,
It has been for me a great honour, as well as a pleasantly challenging activity, to review the article entitled “Participatory design for sustainable living supporting experiential mutual learning, revival of indigenous values, and community resiliency in Sabrang Village Indonesia.”
Overall, the article is interesting. However, I suggest that the Authors introduce a few corrections (given below).
In my opinion, the Introduction chapter well introduces potential readers to the topics discussed by the Authors. The aim of the paper is clearly stated (lines 104-116), however, I would propose a more emphatic emphasis on what is the novelty of this research, what the new article brings to the literature and what research gap it fills.
In addition, I propose to extend the literature review with threads related to examples of ethnographic research conducted in rural areas in other countries on the preservation of cultural heritage, traditional lifestyles, customs, etc., which are widely discussed in the literature, e.g..:
http://doi.org/10.15544/RD.2017.164
https://doi.org/10.3390/su14084858
https://doi.org/10.3390/su13042331
https://doi.org/10.1080/1226508X.2017.1393724
Unfortunately, there are many technical/editorial errors in the work (in this and subsequent chapters), e.g. the text includes references to literature that are not included in the bibliographic list, e.g..: Mulyana, 2014; Wilonoyudho et al., 2017; Hidayat et al., 2020; Pandur, 2017; Geertz, n.d., Feintrenie, 2010; Squires & Tabor, 1991; and many others.
Methods
This chapter is divided into three logically following subchapters and is clearly written. It is well illustrated by four figures, one of which contains a map that allows a potential foreign reader to get a better idea of where the research was carried out. However, the mistakes from the previous chapter are repeated again. There are references to literature that are not included in the bibliographic list and it is difficult to find out as to the legitimacy of quoting them, and it does not make it easier to assess the validity of the research methods used.
Results and discussion
This chapter is broken down into three logically consecutive subchapters. It is well illustrated by two tables and two figures.
However, in my opinion, there is no discussion, i.e. reference to the results of research presented by other scholars. Moreover, I would suggest separating a separate chapter of Conclusions. It would also be valuable to describe the limitations that the Authors encountered while conducting the presented research and to indicate how they could affect the results, their interpretation and the conclusions drawn.
I don't feel competent to comment on linguistic correctness as English is not my mother tongue.
Summing up, in my opinion the manuscript has the potential to attract readers' interest, however, in its current form it is not ready for publication and requires extensive improvement. I wish the Authors good luck!
Round 2
Reviewer 1 Report
The manuscript has been much approved since the last version. Most comments are taken up by the authors. However, the description of the sampling methods (e.g. why this case, why these participants, how were they selected, etc.), and data analysis methods (e.g. how were the interviews analysed, what method was followed to analyse pairwise ranking or likertscale scoring) are still missing in the improved manuscript.
Furthermore, the entire description of the followed research methods (section 3.2) lacks references . Including references in this section would reveal on which scientific tradition and sources the followed methodological approach and applied methods built. This would significantly improve the scientific soundness of the paper, which is very low in the current version. Insights in the followed methods and methodology allows to check the rigor of the research, and, as such, the scientific soundness.
Finally, the response to the comments was too brief and cryptic. A more content related and more elaborated response would have provided insights in how the given comments were taken up, or why certain suggestions were not followed. Furthermore, it was not always clear which parts of the text were not changed, and which were added, removed or significantly altered.
Reviewer 2 Report
Dear Authors,
The references still look a little confusing. Please refer to the format requirements of the journal and revise carefully. It is best to use the MDPI Style in Endnote. Besides, there are too many references in the paper. It is recommended to only keep the most needed ones, and delete some unnecessary ones to prevent them from occupying too much space.
Author Response
Point 1: The references still look a little confusing. Please refer to the format requirements of the journal and revise carefully. It is best to use the MDPI Style in Endnote. Besides, there are too many references in the paper. It is recommended to only keep the most needed ones, and delete some unnecessary ones to prevent them from occupying too much space.
Response 1: Thank you for your comments. We have gone through the manuscript and updated the style to numbered as used by MDPI. You may find the new Bibliography in line 767.
Reviewer 3 Report
I would like to thank the Authors for the corrections introduced and the answers provided. Good luck!
Author Response
I would like to thank the Authors for the corrections introduced and the answers provided. Good luck!
response:
Thank you very much for your review and comments, and also your well-wishing.